# Short-term and long-term cost-effectiveness of a pedometer-based exercise intervention in primary care: a within-trial analysis and beyond-trial modelling

Nana Anokye,[1] Julia Fox-Rushby,[2] Sabina Sanghera,[3] Derek G Cook,[4] Elizabeth Limb,[4] Cheryl Furness,[4] Sally Margaret Kerry,[5] Christina R Victor,[6] Steve Iliffe,[4,7] Michael Ussher,[4] Peter H Whincup,[4] Ulf Ekelund,[8,9] Stephen deWilde,[4] Tess Harris[4]

For numbered affiliations see end of article.

**Correspondence to**
Professor Julia Fox-Rushby;
julia.fox-rushby@kcl.ac.uk

## ABSTRACT

**Objectives** A short-term and long-term cost-effectiveness analysis (CEA) of two pedometer-based walking interventions compared with usual care.

**Design** (A) Short-term CEA: parallel three-arm cluster randomised trial randomised by household. (B) Long-term CEA: Markov decision model.

**Setting** Seven primary care practices in South London, UK.

**Participants** (A) Short-term CEA: 1023 people (922 households) aged 45–75 years without physical activity (PA) contraindications. (b) Long-term CEA: a cohort of 100 000 people aged 59–88 years.

**Interventions** Pedometers, 12-week walking programmes and PA diaries delivered by post or through three PA consultations with practice nurses.

**Primary and secondary outcome measures** Accelerometer-measured change (baseline to 12 months) in average daily step count and time in 10 min bouts of moderate to vigorous PA (MVPA), and EQ-5D-5L quality-adjusted life-years (QALY).

**Methods** Resource use costs (£2013/2014) from a National Health Service perspective, presented as incremental cost-effectiveness ratios for each outcome over a 1-year and lifetime horizon, with cost-effectiveness acceptability curves and willingness to pay per QALY. Deterministic and probabilistic sensitivity analyses evaluate uncertainty.

**Results** (A) Short-term CEA: At 12 months, incremental cost was £3.61 (£109)/min in ≥10 min MVPA bouts for nurse support compared with control (postal group). At £20 000/QALY, the postal group had a 50% chance of being cost saving compared with control. (B) Long-term CEA: The postal group had more QALYs (+759 QALYs, 95% CI 400 to 1247) and lower costs (−£11 million, 95% CI −12 to −10) than control and nurse groups, resulting in an incremental net monetary benefit of £26 million per 100 000 population. Results were sensitive to reporting serious adverse events, excluding health service use, and including all participant costs.

## Strengths and limitations of this study

► This study provides the first primary data on the short-term costs associated with delivering pedometers to a large (n=1023), population-based sample from primary care alongside a high-quality randomised controlled trial that achieved a 93% follow-up rate at 12 months.

► Results from the trial are fed into a peer-reviewed, policy-relevant Markov model to estimate long-term cost-effectiveness as trials of public health interventions are unable to reflect the balance of costs and effects when benefits occur in the long term.

► Results are tested in a number of sensitivity analyses to assess the impact of changing perspective, missing data, changed assumptions about maintenance of physical activity (PA) and of taking more conservative views of outcomes and cost impact.

► The main limitation of the economic analysis is the lack of information about the likelihood of maintaining PA beyond 3 years into the long term and the exclusion of long-term impacts on other conditions, for example, cancers.

**Conclusions** Postal delivery of a pedometer intervention in primary care is cost-effective long term and has a 50% chance of being cost-effective, through resource savings, within 1 year. Further research should ascertain maintenance of the higher levels of PA, and its impact on quality of life and health service use.

**Trial registration number** ISRCTN98538934; Pre-results.

## INTRODUCTION

Increasing physical activity (PA) is a widely stated policy aim from local to international level.[1 2] Walking is a safe and, potentially cheap, activity that has the potential to reduce cardiovascular disease (CVD),

diabetes, cancer and poor mental health.[3] It is therefore important to establish which approaches are effective at: encouraging inactive people to do at least some walking; increasing the number of people walking briskly for at least 150 min/week (ie, achieving moderate to vigorous PA (MVPA) guidelines[2]); and/or maintaining increases in walking over time. This would also provide the basis for estimating cost-effectiveness and supporting recommendations for policy and practice.

Until recently, the best evidence of pedometer-based walking programmes was from systematic reviews that relied on small, short-term studies where the independence of pedometer effects from other support provided was unclear.[4] These had shown that walking interventions can achieve increases of ~2000–2500 steps/day at 3 months, but often relied on volunteer samples or high-risk groups and did not assess time in MVPA, as defined in PA guidelines, as an outcome. New evidence from a large randomised trial clustered by household (Pedometer and Consultation Evaluation-UP, PACE-UP) compared delivery of pedometers by post or through primary care nurse-supported PA consultations. The trial was undertaken with 1023 inactive primary care patients aged 45–75 years from seven practices in South London. Results showed that step counts increased by around 10% and time in MVPA in 10 min bouts by around a third, with both the nurse and postal delivery arms achieving similar 12-month outcomes.[4] This is important because primary care can be a key to reaching directly into the community and offering continuity of care for increasing PA. It is shown that this type of intervention is suitable for older adults, where exercise referral schemes have been disappointing.[4] Compared with national averages (from Health Survey for England 2012 data set) for the same age range of the PACE-UP trial, the trial sample was more overweight/obese (66% vs 61%), more likely to have/ have had a higher managerial, administrative, professional occupation (59% vs 36%) and less likely to be white (80% vs 93%).

Other than a small, highly selected study which limited outcomes to steps achieved among 79 people from one family physician practice in Glasgow,[5] there is no primary evidence of the cost-effectiveness of pedometer programmes in the UK. Elsewhere, in Australia, New Zealand and the Netherlands, economic models from community-based adults with low PA levels compare pedometer prescriptions and pedometer-based telephone coaching with usual practice.[6–8] These indicate pedometer-based interventions may be cost-effective in the long term, but estimates vary widely and generalisability is not considered.[9]

The analytic horizon of cost-effectiveness analyses should extend far enough into the future to capture all benefits and harms, although in practice this can be limited by the amount and quality of data.[10] The National Institute for Health and Care Excellence's (NICE) public health guidance[11] also recommends providing results that reflect the short term (1–3 years). This is reinforced

in NICE's return on investment models,[12] which argue that shorter term decision-making is of key interest to some decision-makers and which have been used by commissioners.

This paper estimates the short-term (1 year) and long-term (lifetime) cost-effectiveness of pedometers delivered by post or through practice nurse consultation for 1023 inactive adults aged 45–75 years. The short-term evaluation arises from a within-trial analysis of individual resource use and costs of interventions provided in the PACE-UP trial.[4] The cost and effectiveness results from the trial are used to populate a long-term model[13] for lifetime cost-effectiveness.

## METHODS
### Short-term cost-effectiveness
The short-term within-trial cost-effectiveness analysis was conducted alongside the PACE-UP trial[4 14] that evaluated two intervention groups against control (no intervention group). The two intervention groups received pedometers (SW-200 Yamax Digi-Walker) (one by post), patient handbook; PA diary (including individual 12-week walking plan), with the nurse group also offered three individually tailored practice nurse PA (10–20 min) consultations (nurse support group only) at approximately weeks 1, 5 and 9.[4] The control group followed usual practice and were not provided with any feedback on their PA levels or materials promoting PA during the trial.[4] These interventions could therefore evaluate the incremental effect of adding nurse support to pedometers.

The costs for the two intervention arms include set-up costs, staff training and intervention delivery (including: pedometers and clips, batteries, handbooks, diaries, postage, nurse time, time making appointments). Measures of each resource use were taken from administrative/trial management records, computer-based diaries and interviews with the trial manager and principal investigator. To account for potential changes in falls, change in use of health services following differential contact of health services by participants or unintended resource consequences, general health service use (eg, general (family) physician visits, hospital admissions, accident and emergency attendances, referrals) was collected at participant level, through a one-time download of physician records at the end of the trial, and linked to procedure codes using PI judgement (blind to treatment group) to facilitate costing across elective and non-elective admissions. Information on costs borne by patients (eg, time use, out-of-pocket expenses associated with walking groups, plus any related travel costs) was collected by questionnaire at 3 and 12 months. Resources were valued using national tariffs where possible[15 16] to increase generalisability; where not available tariffs from St Georges Hospital, London, were used. All costs are expressed in £2013–2014 sterling, inflated to this base year where appropriate using the Hospital & Community Health Service inflation index. As the trial lasted for

1 year, a discount rate was not applied (see online supplementary tables S1–S5). PA was measured objectively by accelerometry (GT3X+, ActiGraph) and data were reduced using ActiLife software (V.6.6.0). The summary variables used were as follows: step counts; and time spent in MVPA in ≥10 min bouts (≥1952 counts/min, equivalent to ≥3 metabolic equivalents).[17]

Outcomes were: (A) changes in daily steps and weekly minutes of MVPA in bouts of ≥10 min, and (B) changes in quality-adjusted life-years (QALY), based on participant completion of the EQ-5D-5L questionnaires at baseline, and 3 and 12 months. Utility weights were assigned using the 'crosswalk' function[18] linked to the standard UK-based weights,[19] with QALYs based on the area under the curve.

Standard practice for accounting for missing data was followed.[20 21] Patterns of missing data were investigated, with multiple imputation by chained equations fitted to replace item non-response. Missing EQ-5D data were replaced using an index rather than domain imputation as recommended.[22] Mean imputation was used where missing data were ≤5%.[23] Imputation models were fitted to match the model used for main analysis while including the predictors of missingness as appropriate. Second, the dependent variables were included in imputation models to ensure that the imputed values have similar relationships to the dependent variable as the observed values.[24]

Results are reported, from a National Health Service (NHS) perspective, as incremental cost-effectiveness ratios (ICER) for cost per change in daily steps and cost per QALY for a 1-year time period, adjusted for baseline differences. A generalised linear model was fitted separately for costs and QALYs with clustered SEs. To provide more precise estimates of uncertainty, the 'margins method' was used to generate sample means by trial arm for costs and QALYs.[24] Cost models were fitted using the Poisson distribution and QALY models using the binomial 1 family, equivalent to beta regression.[25] The choice of distributional family for the models was based on the modified Park test and comparison of observed and predicted values. Covariates included baseline level (for the QALY-based models),[26] practice and variables found to be correlates of PA-related outcomes,[27] that is, demography (age, gender, ethnicity, marital status, education, employment, socioeconomic status, cohabitation), health (number of disease conditions) and other lifestyle behaviours (smoking and alcohol intake). Reduced models were generated using Wald tests to examine the joint significance of variables found not to be significant (at 5%) in the base model.

Deterministic sensitivity analyses assessed: (A) inclusion of all randomised patients (rather than only those who provided accelerometry data); (B) exclusion of costs of general health service use beyond immediate intervention; (C) exclusion of missing data; (D) methods of accounting for adverse events; (E) perspective of analysis (ie, including all and parts of participant costs); (F) varying the length of life of a pedometer; (G) the combination of excluding all health service use costs; and (H)

including participant costs related to participation in PA and the interventions (minus health service use cost borne by participants, to ensure consistency in perspective). To reflect stochastic uncertainty surrounding mean incremental cost-effectiveness, cost-effectiveness planes and cost-effectiveness acceptability curves (CEAC) were constructed using 2000 non-parametric bootstrap samples from the base case estimates.

## Long-term cost-effectiveness

A Markov model used to support NICE public health guidance[28] and return on investment modelling[12] was adapted to examine the long-term (lifetime) cost-effectiveness. From an NHS perspective, costs (2013/2014 prices) and health outcomes from reduced disease, expressed as QALYs, were discounted at the rate of 3.5% per annum. Results are reported as ICER, CEAC and incremental net benefit statistics.

In the original model,[13] a cohort of 100 000 people aged 33 years were followed in annual cycles over their lifetime. At the end of the first year of the model, the cohort is either 'active' (doing 150 min of MVPA in 10 min bouts per week) or 'inactive' and they could have one of three events (non-fatal coronary heart disease (CHD), non-fatal stroke, type 2 diabetes (T2D)), remain event free (ie, without CHD, stroke or diabetes) or die either from CVD or non-CVD causes, each of which had assigned annual treatment costs (split by initial event and follow-up). After the first year, people would revert to PA patterns observed in long-term cohort studies (up to 10-year cycle in the model) on the relationship between PA and disease conditions.[13] The key driver of the long-term model is the protective effects of PA, which is a function of PA patterns after the first year of the intervention. In the base case analysis, PA behaviour was based on PA patterns observed in long-term cohort studies[29–31] on the relationship between PA and disease conditions. The cohort studies used followed up the same people (who were either active or inactive at baseline) for 10 years, during which some of the inactive people might have become active or vice versa. Thus, the impact of changing habits is incorporated in the cohort relative risk (RR) estimates from these epidemiological studies. However, assuming that these estimates would persist after the follow-up periods might be impractical. It was therefore assumed, conservatively, that these RR estimates held for an initial 10-year period (ie, the period PA patterns were observed in the epidemiological studies), after which no protective benefit would persist. Hence, the RRs for developing CHD, stroke and T2D in the first 10 years of the model were based on the estimates from the epidemiological studies but from year 11 onwards they were assumed to be equal to 1 (no effect). This assumption was tested for sensitivity analyses.

Active individuals had lower risks of developing CHD, stroke and T2D. People who become active in the first year (irrespective of trial arm) also accrue short-term psychological benefits, a one-off utility gain associated

with achieving the recommended level of PA[13] (see online supplementary figure S1).

The model was adapted using data from the PACE-UP trial in the following ways: (A) a cohort of 100 000 people aged 59 years followed, in annual cycles, to 88 years, reflecting the average age of all trial participants at baseline and the average life expectancy for people aged 59 years in the UK[32] and exposed, at this age, to interventions (either nurse or postal) in an unexposed population, that is, control group/usual care:

a. Age-specific estimates were revised to reflect the change in the cohort age.
b. Within-trial cost of interventions was used, with a second year of annuitised values included appropriately—postal (£5.03/person) and nurse group (£4.14/person).
c. Effectiveness was reflected as the RR of achieving ≥150 MVPA min/week in ≥10 min bouts.
d. Short-term psychological benefits of PA (one-off utility gain) estimated using beta regression fitted for EQ-5D scores at 12 months for active people controlling for EQ-5D scores at baseline, demographics, practice, disability and trial arm using.

All other parameters remained the same as the original model, based on literature reviews or evidence from national/international science-based guidance on PA and health. Parameter estimates are provided in online supplementary table S6.

Deterministic sensitivity analysis explored four conservative scenarios: (1) Assuming the protective effects of PA exist only for 1 year, as the trial MVPA data were assessed at 12 months. (2) Assuming the protective effects of PA exist for 3 years. Recent evidence[33] relating to 3-year follow-up of participants of the interventions showed persistent effect at 3 years. (3) Exclusion of all health service use cost consequences during trial period (model year 1) and assumed no psychological benefits in the first year of being physically active. This was considered due to the uncertainty around short-term changes to health service use and because previous studies found the exclusion of short-term QALY gain associated with being physically active to affect conclusions.[13] (4) Scenario 3 plus all patient costs related to participation in PA and the interventions (details of the participant costs are provided in online supplementary table S4). Probabilistic sensitivity analysis was based on 10 000 Monte Carlo simulations and included all parameters except baseline mortality, as the mortality census data have little uncertainty.

### Patient and public involvement

Patient and public involvement across the study is described in our publication of the main results[4] and is reproduced below under the terms of the Creative Commons Attribution Licence (CC BY 4.0).

Pilot work with older primary care patients from three general practices was carried out ahead of seeking trial funding, with focus groups at each practice discussing ideas for a pedometer-based PA intervention. Patients were enthusiastic about the study and felt that the postal approach to recruitment and the interventions offered would be acceptable. They had input into aspects of the study design; for example, they encouraged us to offer the usual care arm a pedometer at the end of the follow-up period and they encouraged us to recruit couples as well as individuals, and to allow couples to attend nurse appointments together.

A patient advisor was a Trial Steering Committee member and was involved in discussions about recruitment and study conduct, as well as advising about patient materials, dissemination of results to participants and safety reporting mechanisms.

All participants were provided with timely feedback of their individual trial results after completion of 12-month follow-up, including their PA and body size measures over the trial duration. Summaries of results for the whole trial were disseminated to all trial participants as A4 feedback sheets after completion of baseline assessments and after analysis of the main results. A trial website (http://www.paceup.sgul.ac.uk/) has been created, and details have been circulated to participants. This also provides a summary of the trial results and details about further trial follow-up. All publications relating to the trial are provided on the website.

The burden of the intervention was assessed by all participants in the nurse group with a questionnaire as part of the process evaluation[34] and by samples of both intervention groups as part of the qualitative evaluation.[35]

## RESULTS

### Short-term cost-effectiveness

Table 1 summarises data on costs, EQ-5D-5L utility scores and QALYs by trial arm. At 3 months, average cost per participant was highest in the nurse group (£249) followed by the postal (£122) and control group (£107). In terms of the components of total costs, the cost of nurse-supported pedometer delivery was seven times greater (£50) than the postal group (£7), and set-up costs were double. Comparing the trial arms based on cost of health service use shows that the control group costs £35 more per participant than the postal group and £12 more than the nurse group. Results are similar at 12 months, except for the control arm, which has a higher overall average cost than the postal arm.

Table 2 shows that, at 3 months, mean incremental costs were significantly higher for the nurse group compared with the postal (+£120, 95% CI £95 to £146) and control groups (+£135, 95% CI £99 to £171) but not statistically significantly higher for the postal compared with control group. While increases in both daily steps and weekly minutes of MVPA in ≥10 min bouts for both interventions compared with control, and for the nurse group compared with postal (nurse: +481 steps (95% CI 153 to 809), +18 min MVPA (95% CI 1 to 35)) were statistically significant, the small mean decrease in QALYs is not statistically significant for any comparison. The cost per additional minute

**Table 1** Average costs and QALYs per participant, by trial arm (£2013/2014 sterling, all randomised participants who provided required accelerometry data*, missing data imputed)

| Cost and quality of life (EQ-5D-5L) | Control | Postal† | Nurse† |
|---|---|---|---|
| | **Mean (SD)** | | |
| **0–3 months** | **n=318** | **n=317** | **n=319** |
| Total cost | £107 (254) | £122 (107) | £249 (215) |
| Set-up | £0 (0) | £45 (0) | £105 (0) |
| Delivery of intervention | £0 (0) | £7 (0) | £50 (18) |
| Health service use | £107 (254) | £71 (107) | £95 (214) |
| EQ-5D scores at baseline | 0.839 (0.14) | 0.853 (0.12) | 0.851 (0.12) |
| EQ-5D scores at 3 months | 0.844 (0.14) | 0.848 (0.14) | 0.841 (0.14) |
| QALYs 0–3 months | 0.194 (0.03) | 0.196 (0.03) | 0.195 (0.03) |
| **0–12 months** | **n=323** | **n=312** | **n=321** |
| Total cost | £461 (916) | £375 (611) | £603 (987) |
| Set-up | £0 (0) | £45 (0) | £105 (0) |
| Delivery of intervention | £0 (0) | £10 (0) | £52 (18) |
| Health service use | £461 (916) | £320 (611) | £447 (987) |
| EQ-5D scores at baseline | 0.837 (0.14) | 0.850 (0.12) | 0.849 (0.13) |
| EQ-5D scores at 3 months | 0.840 (0.14) | 0.847 (0.13) | 0.837 (0.14) |
| EQ 5D scores at 12 months | 0.833 (0.15) | 0.836 (0.13) | 0.831 (0.14) |
| QALYs 0–12 months | 0.837 (0.13) | 0.843 (0.11) | 0.836 (0.13) |

*The number of people who provided accelerometry data differed across time points within arms.
†For incremental analyses, the comparisons are postal versus control and nurse versus control.
QALY, quality-adjusted life-year.

of MVPA was 35p for postal group and £2.21 for the nurse group and therefore the (slightly) fewer QALYs for both interventions compared with control contributed to the dominance of each intervention by the control group (ie, the control group cost less and had more QALYs). To move from a postal to nurse-delivered pedometer would cost 25p per additional step and £6.67 per additional MVPA-min. However, in terms of cost-effectiveness, the nurse group costs more and produces less QALYs on average than the postal group at 3 months.

Results differ at 12 months. Compared with the control group, the postal arm cost less on average (−£91) and the nurse group more (+£126) but neither are statistically significant. The increase in cost of moving from a postal to nurse delivery is also statistically significantly higher (+£217, 95% CI £81 to £354). While both interventions are associated with a statistically significant increase in steps and weekly minutes of MVPA, the difference between intervention groups is not statistically significant at 12 months. The small decrements in QALYs at each incremental comparison are not statistically different. The postal group took more steps (+642) and cost less on average (−£91) compared with control and dominates control in terms of PA outcomes. The nurse group cost 19p per additional step and £3.61 per additional minute of MVPA compared with control, with this rising to £6 and £109, respectively, when compared with the postal group. In terms of QALYs, the nurse group is still dominated (ie,

cost more and had worse outcomes) by the control and postal groups. However, on average, each QALY lost in the postal group compared with control is associated with a saving of £21 162, which could therefore be considered cost-effective.

The probabilistic sensitivity analyses broadly confirm the findings of the base case; the postal group is most often associated with lower QALYs along with cost savings and the nurse group tends to have both lower QALYs and higher costs compared with control and postal group (online supplementary tables S2–S4 and supplementary figures S3 and S4). Figure 1 shows that at £20 000 per QALY gained/lost, the postal group has a 50% chance of being cost-effective compared with control (usual care). This falls to 42% at £30 000/QALY, which reflects the postal group having most observations in the lower left-hand quadrant (as seen in online supplementary figure S2). Figure 1 also shows that, at a willingness to pay/lose a QALY of £20 000, the nurse group has a 5.5% chance of being cost-effective compared with control.

The deterministic sensitivity analyses (online supplementary table S7) mostly produced results consistent with the base case findings. However, in four circumstances, usual care would dominate both the postal and nurse groups at 12 months: (1) using health service use based on self-reported serious adverse effects; (2) excluding all health service costs; (3) changing perspective (including

**Table 2** Regression estimates for costs, effects and cost-effectiveness at 3 and 12 months (£2013/2014 sterling) (base case, adjusted for baseline differences)

| Cost, effects or cost-effectiveness | Control | | Postal* | | Nurse* | | Nurse versus postal | |
|---|---|---|---|---|---|---|---|---|
| | Mean | (95% CI) | Mean | (95% CI) | Mean | (95% CI) | Mean | (95 % CI) |
| **Costs and effects over 3 months** | | | | | | | | |
| Total costs per participant (£) | 108 | (80 to 136) | 123 | (111 to 135) | 244 | (221 to 266) | – | – |
| Incremental cost (£) | – | | 15 | (–15 to 45) | 135 | (99 to 171) | 120 | (95 to 146) |
| Total QALYs per participant | 0.1957 | (0.1936 to 0.1978) | 0.1952 | (0.1930 to 0.1974) | 0.1948 | (0.1926 to 0.1970) | – | – |
| Incremental* QALYs | – | | –0.0005 | (–0.0027 to 0.0016) | –0.0009 | (–0.0031 to 0.0012) | –0.0004 | (–0.0026 to 0.0018) |
| Incremental daily steps | | | 692 | (363 to 1020) | 1172 | (844 to 1501) | 481 | (153 to 809) |
| Incremental weekly minutes of MVPA in bouts of ≥10 min | | | 43 | (26 to 60) | 61 | (44 to 78) | 18 | (1 to 35) |
| **Costs and effects over 12 months** | | | | | | | | |
| Total cost per participant (£) | 467 | (365 to 569) | 376 | (307 to 445) | 593 | (473 to 714) | – | |
| Incremental cost (£) | – | | –91 | (–215 to 33) | 126 | (–37 to 290) | 217 | (81 to 354) |
| Total QALYs per participant | 0.842 | (0.832 to 0.853) | 0.838 | (0.827 to 0.849) | 0.836 | (0.824 to 0.847) | – | |
| Incremental QALYs | – | | –0.004 | (–0.017 to 0.009) | –0.007 | (–0.020 to 0.007) | –0.002 | (–0.016 to 0.011) |
| Incremental daily steps | – | | 642 | (329 to 955) | 677 | (365 to 989) | 36 | (–227 to 349) |
| Incremental weekly minutes of MVPA in bouts of ≥10 min | – | | 33 | (17 to 49) | 35 | (19 to 51) | 2 | (–14 to 17) |
| **ICER* at 3 months** | | | | | | | | |
| Cost per additional QALY (£) | | | Postal dominated by control | | Nurse dominated by control | | Nurse dominated by postal | |
| Cost per additional step count (£) | – | | £0.02 | | £0.12 | | £0.25 | |
| Cost per additional minute of MVPA in a bout of ≥10min (£) | £0.35 | | £0.35 | | £2.21 | | £6.67 | |
| **ICER* at 12 months** | | | | | | | | |
| Cost per additional QALY (£) | – | | Postal is less costly but has fewer QALYs. £211 162 saved per QALY lost. | | Nurse dominated by control | | Nurse dominated by postal | |
| Cost per additional step count (£) | – | | Postal dominates control | | 0.19 | | 6.03 | |
| Cost per additional minute of MVPA in a bout of ≥10min (£) | – | | Postal dominates control | | 3.61 | | 109.00 | |

ICER, incremental cost-effectiveness ratio; MVPA, moderate to vigorous physical activity; QALY, quality-adjusted life-year; *versus control.

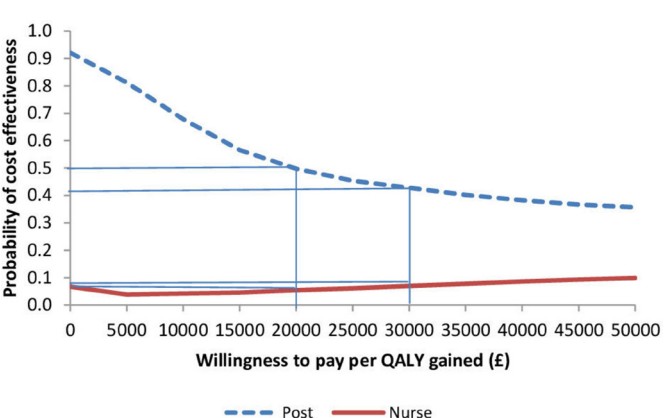

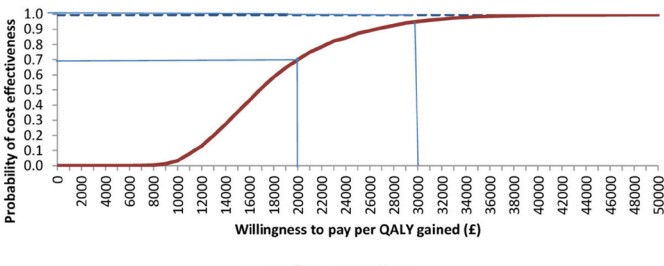

**Figure 2** Cost-effectiveness acceptability curve showing the probability of lifetime cost-effectiveness for postal and nurse groups (vs control) at different willingness to pay per quality-adjusted life-year (QALY) threshold levels.

**Figure 1** Cost-effectiveness acceptability curve showing the probability of short-term (at 1 year) cost-effectiveness for postal and nurse groups (vs control) at different willingness to pay per quality-adjusted life-year (QALY) thresholds.

all participant costs); and (4) the worst case 'combined scenario' sensitivity analyses.

### Long-term cost-effectiveness

Table 3 shows that, over the remaining lifetime from age 59, the nurse group would be costlier (£11 million, 95% CI £10 million to £12 million) but have more QALYs (671, 95% CI 346 to 1071) per 100 000 population than the control group and therefore provide each additional QALY at a cost of £16 368. However, the postal group would have lower lifetime costs than the control arm (−£11 million per 100 000 population, 95% CI −£12 million to −£10 million) and more QALYs (759, 95% CI 400 to 1247); it is therefore the dominant option, with an incremental net benefit of £26 million per 100 000 population (95% CI £18 million to £36 million). These results are confirmed by the incremental net benefit, which shows the £2 million per 100 000

for nurse group compared with control is not significantly different and compared with the post group is significantly negative (−£24 million, 95% CI −£27 to −£21).

The stochastic uncertainty associated with the mean ICER (figure 2) indicates the above findings are robust. There is a 100% likelihood, at a willingness to pay £20 000/QALY, that the postal group is cost-effective compared with the control and nurse groups. This is consistent with the estimates of net monetary benefit in table 3. At £20 000/QALY, there is a 70% likelihood that the nurse group would be cost-effective compared with control (figure 2).

The results for the sensitivity analyses were:

a. Scenario 1: (1) postal versus control: postal remained dominant, less expensive (−£9 million) with more QALY gains (+211 QALYs); (2) nurse versus control: the ICER further increased from £16 000 to £69 000 (+£12.8 million, +186 QALYs); (3) nurse versus postal: the nurse group remained dominated by postal group (+£21.6 million, −32 QALYs).

**Table 3** Costs, effects and cost-effectiveness over a lifetime from age 59 (cohort of 100 000 people)

| | Control | Postal* | Nurse* | Nurse versus postal |
|---|---|---|---|---|
| | Mean (95% CI) | Mean (95% CI) | Mean (95% CI) | Mean (95% CI) |
| Lifetime total cost (£ million)† | 340 (307 to 371) | 329 (296 to 361) | 351 (318 to 384) | – |
| Lifetime incremental cost (£ million) | – | −11 (−12 to 10) | 11 (10 to 12) | 22 (21 to 23) |
| Lifetime total QALYs (million) | 1.0709 (0.879 to 1.273) | 1.0717 (0.889 to 1.274) | 1.0716 (0.880 to 1.273) | – |
| Lifetime incremental QALYs | – | 759 (400 to 1247) | 671 (346 to 1071) | −108 (−223 to −10) |
| Lifetime ICER for QALYs (£) | | Postal dominates control | 16 368 | Postal dominates nurse |
| Lifetime incremental net monetary benefit (£ million, @£20 000 per QALY) | – | 26 (18 to 36) | 2 (−5 to 11) | −24 (−27 to −21) |

*For incremental analyses, the comparisons are postal versus control and nurse versus control.
†£46.7 million, £37.6 million and £59.3 million of the total costs for control, postal and nurse groups, respectively, were estimated using Pedometer and Consultation Evaluation-UP (PACE-UP) trial results.
ICER, incremental cost-effectiveness ratio; QALY, quality-adjusted life-year.

b. Scenario 2: (1) postal versus control: postal was still dominant, less expensive (−£9.2 million) with more QALY gains (+327 QALYs); (2) nurse versus control: the ICER increased from £16 000 to £43 000 (+£12.4 million, +289 QALYs); (3) nurse versus postal: the nurse group remained dominated by postal group (+£21.7 million, −48 QALYs).

c. Scenario 3: (1) postal versus control: postal moved from a dominant position to a more expensive option (+£4 million) with more QALY gains (+609 QALYs), and an ICER of £6100; (2) nurse versus control: the ICER increased from £16 000 to £26 000 (+£14 million, +538 QALYs); (3) nurse versus postal: the nurse group remained dominated by postal group (+£10 million, −87 QALYs).

d. Scenario 4: (1) postal versus control: postal moved from a dominant position to more expensive (+£16 million) and more QALY gains (+609 QALYs) with an ICER of £26 600; (2) nurse versus control: the ICER increased from £16 000 to £25 400 (+£13.7 million, +538 QALYs); (3) nurse versus postal: nurse moved from dominated position (where costs are higher and QALYs lower) to a cost-effective position (where both costs and QALYs are lower) (−£2 million, −87 QALYs).

## DISCUSSION

The lifetime cost-effectiveness of posting a pedometer with written instructions to a cohort of 100 000 insufficiently active people aged 59 years (who have indicated an interest in research or participation in walking) would cost less (−£11 million, 95% CI −12 to −10) and provide more QALYs (759 QALYs, 95% CI 400 to 1247) than usual care. Most cost savings and quality of life benefits derive from reductions in stroke, CHD and T2D. This finding was robust (incremental net benefit of £26 million, 95% CI £18 million to £36 million) and sensitivity analyses showed that even excluding short-term cost savings would not change the conclusion that the postal group would be extremely cost-effective in the long term (ICER: £6100/QALY). Sending a pedometer by post with instructions from a primary care provider to inactive people aged 45–75 also has a 50% chance of being cost-effective *within a year*, as a 1 QALY loss was associated with saving over £21 000. The nurse group had higher costs and lower QALYs than both control and postal groups at 1 year. While sensitivity analyses did not change conclusions in most cases, in three cases (using self-reported serious adverse events, excluding health service use, including all participant costs) it did, indicating that the control group would dominate (ie, have lower costs and more QALYs) than both the postal and nurse groups.

A key strength of this study is the base of individualised cost and effectiveness data on a large, population-based, cluster randomised controlled trial with excellent follow-up data to 1 year (93.4%, Harris *et al*),[4] designed to produce generalisable results, for cost per QALY estimates at 1 year and as inputs to a long-term model of cost-effectiveness. It is also the only study to have included provider and user perspectives, extended commonly used techniques to account for clustering and used conservative assumptions for both short-term and long-term sensitivity analyses.

One weakness of the within-trial cost-effectiveness study concerns the use of PI judgement to determine costs of admissions, and therefore alternative assumptions were explored in sensitivity analyses. Patient-reported cost data were collected for months 1–3 and 9–12, with the last 3 months multiplied to represent costs across all months from 4 to 12. If significantly underestimated, this could be decisional. To date, there are no primary economic data beyond 12 months of an intervention and very few trials include measures of quality of life alongside PA. Therefore, with respect to the long-term modelling, a key gap in knowledge is the likelihood of maintaining PA beyond 12 months. This model assumes differences in PA at 1 year in the trial relate to the same long-term benefit associated with the same difference in cohort studies, but this could be updated once longer term follow-up data become available. Other challenges set out in Anokye *et al*'s[13] study are relevant here, for example, cancer and adverse events are not accounted for, which could lead to overestimation or underestimation of cost-effectiveness. Other challenges relate to the generalisability of effectiveness data, given the focus on South London and 10% recruitment rate, even though recruitment was comparable with other PA trials.[36 37] The trial was shown to recruit fewer: men, people aged 55–64 years compared with those over 65 years, people from the most deprived quintile compared with least deprived and Asian compared with white people.[37] However, there was good representation of women, older adults and people who were overweight, all of whom are groups likely to benefit from the intervention.[4] Investigation into the reasons for non-participation showed an important minority cited existing medical conditions, too many other commitments or considered themselves sufficiently active.[35 38]

This study feeds into an area with very limited primary data[39 40] populated only by small studies.[5 6] In New Zealand, pedometers were shown to have a 95% probability of being a cost-effective addition to green prescriptions at 12 months,[5] much higher than the 50% likelihood we found. Other models of long-term cost-effectiveness studies identified cost savings and improved quality of life at a population level from pedometers in the long term[8 41] or indicated high probabilities of long-term cost.[7 42] Guidance has also suggested that long-term monitoring/support at £25/year would be very cost-effective. Our study provides further support that pedometer-based programmes are a cost-effective method of improving health-related quality of life in both the short term and long term. Assumptions about intervention effectiveness beyond 1 year have mixed impacts, and further research is required to better judge whether existing models overpredict or underpredict cost-effectiveness.

Current public health guidance from NICE on pedometers[43] advises using pedometers as 'part of a package' which includes support to set realistic goals in one-to-one meetings (where the number of steps taken is gradually increased), monitoring and feedback. Our results provide substantially better economic data for use by NICE and suggest guidance should be updated to reflect the value of providing pedometers to people who have made some form of commitment (ie, to a trial) through the post. For those practices that have implemented consultation-based distribution of pedometers, moving to postal delivery could save costs within a year, with similar outcomes.

Postal delivery of pedometer interventions to inactive people aged 45–75 through primary care is cost-effective in the long term and has a 50% chance of being cost-effective, through resource savings, within 1 year. Further research is needed to ascertain the extent to which higher PA levels are maintained beyond 3 years and the impact of PA on quality of life and general health service use in both the short term and long term.

**Author affiliations**
[1]Health Economics Research Group, Brunel University, London, UK
[2]Department of Population Health Sciences, Guy's Campus, King's College London, London, UK
[3]School of Social and Community Medicine, University of Bristol, Bristol, UK
[4]Population Health Research Institute, St George's University of London, London, UK
[5]Pragmatic Clinical Trials Unit, Queen Mary's University of London, London, UK
[6]Department of Clinical Sciences, Brunel University London, Uxbridge, UK
[7]Research Department of Primary Care and Population Health, University College London, London, UK
[8]Department of Sport Medicine, Norwegian School of Sport Sciences, Oslo, Norway
[9]MRC Epidemiology Unit, University of Cambridge, Cambridge Biomedical Campus, Cambridge, UK

**Acknowledgements** We thank Iain Carey for help with processing the primary care data as well as Abbie Hill and Julie Whittaker who ably supported the administration of the economics project at Brunel University London. We remember and thank too, our colleague and coinvestigator, Dr SM Shah who died in September 2015. We thank the South-West London (UK) general practices, their practice nurses who supported this study, and all the patients from these practices who participated: Upper Tooting Road Practice, Tooting; Chatfield Practice, Battersea; Wrythe Green Practice, Carshalton; Francis Grove Practice, Wimbledon; Putneymead Practice, Putney; Heathfield Practice, Putney; and Cricket Green Practice, Mitcham. We also thank our supportive Trial Steering Committee: Professor Sarah Lewis (chair); Professor Paul Little (GP representative); Mr Bob Laventure (Patient and Public Involvement representative).

**Contributors** JFR conceived the economic analysis, was coapplicant for funding, jointly designed the economic data collections tools, wrote the economic analysis plan, collected part of the data, supervised the economics and jointly drafted and amended the script. She is the guarantor for this script. NA jointly designed the data collection tools, cleaned and analysed the economic data, and jointly drafted and amended the script. SS collated and analysed the hospital cost data, commented on drafts and reviewed the final script. DGC, EL and SMK designed data collection for and analysed the intermediate outcome data underpinning the economic analyses, discussed plans and results as presented through the trial, commented on drafts of this manuscript and reviewed the final script. CF collated and provided access to the administrative data used for the economic analysis, was the research project administrator, commented on drafts and reviewed the final script. TH was the principal investigator, involved at all points of the planning, progress and review of the economic evaluation including commenting on drafts and review of the final script. She is the guarantor for the whole trial. CRV, PHW, MU, SI, UE and SdW all conceived the trial plan and applied for funding, they contributed to conceptualisation of the economics within the broader context of the trial, discussed plans and results as presented through the trial, commented on drafts of this manuscript and revised the final script.

**Funding** This research was supported by the Health Technology Assessment (HTA) Programme, National Institute for Health Research (project number HTA 10/32/02 ISRCTN42122561).

**Disclaimer** The views and opinions expressed therein are those of the authors and do not necessarily reflect those of the Health Technology Assessment (HTA) Programme, National Institute for Health Research, National Health Service or the Department of Health.

**Competing interests** JFR reports board membership of the Public Health Research Funding Board of NIHR. TH sits on Primary Care Community Interventions Panel for the Health Technology Assessment Programme of NIHR. All authors have been contracted to evaluate other public health interventions, including pedometers, for the NIHR.

**Patient consent** Not required.

**Ethics approval** NHS Ethics Committee.

**Provenance and peer review** Not commissioned; externally peer reviewed.

**Data sharing statement** Data are available upon request from TH.

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
