## [Reviewer comments · BMJ Open]

ARTICLE DETAILS

TITLE (PROVISIONAL)	The short-term and long-term cost-effectiveness of a pedometer-based exercise intervention in primary care: A within-trial analysis and beyond-trial modelling
AUTHORS	Anokye, Nana; Fox-Rushby, Julia; Sanghera, Sabina; Cook, Derek; Limb, Elizabeth; Furness, Cheryl; Kerry, Sally; Victor, Christina; Iliffe, Steve; Ussher, Michael; Whincup, Peter; Ekelund, Ulf; DeWilde, Stephen; Harris, Tess

VERSION 1 – REVIEW

REVIEWER	Sebastian Hinde Centre for Health Economics, University of York, UK
REVIEW RETURNED	01-Mar-2018

GENERAL COMMENTS	Overall a well designed and presented paper, my only two recommendation of any significance would be: A. add a short section of the key drivers of the long-term model, while the results of the trial and how they relate to the short-term analysis are presented in some detail I was not left with a clear picture of what drives the long-term cost-effectiveness. From my read through (and interpretation of the results) it is the MVPA that is driving the result (given the limited length of the short-term model) then a clearer description is needed as to how the MVPA is modelled as having an effect. I know there is a brief description and reference to the published model used but if there is only a single driver of the long-term model this needs to be clear throughout as it is then the key parameter of interest. B. You need to do more to explore the role of your assumption around the continuation of PA, detailed from line 16 p12. You assume a continuation of effect, but that seems like the biggest assumption of all, and yet you don't explore it through sensitivity. What if the impact was for 1/5/10 years, what if it showed exponential decay. Given the small (and statistically insignificant) difference in MVPA at the end of 1 year I think this would massively impact the long-term result, so I'm not clear what it is given such a limited consideration. Again this ties into point A about how much detail is given between the short and long-term model. Minor comments: 1. line 11 of the abstract I assume you mean 45-75 rather than 45+-75?2. The results section of the abstract presents too many results without enough context, I'd suggest taking out some of the cost/step and focussing on the long-term results.3. It might be worth reflecting more on the limited geographic area your analysis covered. How does this area compare nationally in terms of fitness/obesity/wealth levels.
---

	4. line 20 of page 4 I assume you mean 'It is shown' rather than 'It is shows' 5. line 28 p4 you are critical of the generalizability of other studies but do you feel yours is highly generalizable? 6. line 17 p5 'two' rather than 'wo' and 'against a control'. Overall the paper really needs a good proof read, it was a bit disappointing to see so many errors that even a word processor should have picked up. 7. p6 you talk about the missing data but what was the impact of your imputation, I am happy for it to be the base-case but it would be good to know what the scale of missingness was and something of the impact of imputation. Did you follow a good practice guide e.g. Faria et al (2015)? 8. p8 line 3, why not a full lifetime scale rather than to 88 years? Won't make much difference at all 9. p8 line 23, '...physical activity and the interventions.' where what, included/excluded? 10. It isn't very clear if the control group matches what you consider to be a 'no intervention' group, or by the time patients were enrolled are they already subject to some intervention? From a knee jerk reaction their costs seemed high in the short-term model if they are no intervention 11. End of p8 start of p9 you need to be clearer about the costs, I didn't really follow why you would include or exclude the different costs. 12. Was different EQ5D at baseline adjusted for? The change over the year seems so small that I'm not really sure what to make of the results, or is that what you aimed to prove? I cant imagine that any intervention like this that is aiming for a long term impact would have a noticeably one year QoL impact. 13. I don't see why Figure 1 and 2 don't report the control, a CEAC isn't incremental it is raw % probability of being cost-effective at different thresholds for all interventions. Why are you only reporting for 2 interventions? Were these results used for your headline % CE? 14. You should reflect more that the patient group were self-selected, this is very unlikely to anywhere near as effective in a more general population. 15. A brief comment on why the total QALYs in table 3 are the same across groups but incremental are different would be helpful. I assume it is a rounding issue, in which case you might want to report more decimal places. I also find total QALYs easier to interpret if reported as a per patient average but that might be personal preference only.
--	---

REVIEWER	Robert Sloan PhD Kagoshima Graduate Medical School, Japan
REVIEW RETURNED	29-Apr-2018

GENERAL COMMENTS	The analysis must be redone not using > 10minute bouts. The guideline has changed. Please read https://health.gov/paguidelines/second-edition/report/pdf/02_A_Executive_Summary.pdf Bouts, or episodes, of moderate-to-vigorous physical activity of any duration may be included in the daily accumulated total volume of physical activity. The 2008 Physical Activity Guidelines for Americans recommended accumulating moderate-to-vigorous physical activity in bouts of 10 minutes or more. Research now
--

	shows that any amount of moderate-to-vigorous physical activity counts toward meeting the target range. Previously, insufficient evidence was available to support the value of bouts less than 10 minutes in duration. The 2018 Committee was able to conclude that bouts of any length contribute to the health benefits associated with the accumulated volume of physical activity.
--	---

VERSION 1 – AUTHOR RESPONSE

Reviewer: Sebastian Hinde

Reviewer’s comments	How addressed	Location of revision
Major comments		
Overall a well-designed and presented paper, my only two recommendation of any significance would be: A. add a short section of the key drivers of the long-term model, while the results of the trial and how they relate to the short-term analysis are presented in some detail I was not left with a clear picture of what drives the long-term cost-effectiveness. From my read through (and interpretation of the results) if is the MVPA that is driving the result (given the limited length of the short-term model) then a clearer description is needed as to how the MVPA is modelled as having an effect. I know there is a brief description and reference to the published model used but if there is only a single driver of the long-term model this needs to be clear throughout as is then the key parameter of interest. B. You need to do more to explore the role of your assumption around the continuation of PA, detailed from line 16 p12. You assume a continuation of effect, but that seems like the biggest	Thank you for your careful review and kind words. We have added text (see below) to clarify the driver of the model and conducted 2 further additional sensitivity analyses, to explore the impact of assumptions about continuation of PA. Text added to methods section A. The key driver of the long-term model is the protective effects of PA, which is a function of PA patterns after the first year of the intervention. In the base case analysis, PA behaviour was based on PA patterns²⁵⁻²⁷ observed in long-term cohort studies of the relationship between PA and disease conditions. The cohort studies followed up the same people (who were either active or inactive at baseline) for 10 years, during which some of the inactive people might have become active or vice versa. Thus the impact of changing habits is incorporated in the cohort RR estimates from these epidemiological studies. However, assuming that these estimates would persist continuously after the follow-up periods might be questionable. It was therefore assumed, conservatively, that these RR estimates held for an initial 10-year period (i.e. the period PA patterns were observed in the epidemiological studies), after which no protective benefit would persist. Hence, the RRs for developing CHD, stroke and T2D in the first 10 years of the model were based on the estimates from the epidemiological studies but from year 11 onwards they were assumed to be equal to 1 (no effect). This assumption was tested in sensitivity	Page 7-8 (last paragraph of page 7; and 1st paragraph of page 8)

Reviewer's comments	How addressed	Location of revision
assumption of all, and yet you don't explore it through sensitivity. What if the impact was for 1/5/10 years, what if it showed exponential decay. Given the small (and statistically insignificant) difference in MVPA at the end of 1 year I think this would massively impact the long-term result, so I'm not clear what it is given such a limited consideration. Again this ties into point A about how much detail is given between the short and long-term model.	analyses. B. Deterministic sensitivity analysis explored four conservative, scenarios: (1) assuming the protective effects of PA exist only for 1 year, as the trial MVPA data was assessed at 12 months (2) assuming the protective effects of PA exist for 3 years, as recent evidence²⁸ relating to 3 year follow-up of participants of the interventions showed persistent effect at 3 years; (3)..... Text added to results section The results for the sensitivity analyses were: (a) Scenario 1 - (i) postal vs control: postal remained dominant, less expensive (-£9m) with more QALY gains (+211QALYs); (ii) Nurse vs control: The ICER further increased from £16,000 to £69,000 (+£12.8m, +186QALYs); (iii) Nurse vs postal: The Nurse group remained dominated by postal group (+£21.6m, -32QALYs). (b) Scenario 2 - (i) postal vs control: postal was still dominant, less expensive (-£9.2m) with more QALY gains (+327QALYs); (ii) Nurse vs control: The ICER increased from £16,000 to £43,000 (+£12.4m, +289QALYs); (iii) Nurse vs postal: The Nurse group remained dominated by postal group (+£21.7m, -48QALYs).	Page 9 (1st paragraph of page 9) Page 11 (last paragraph)
Minor comments		
line 11 of the abstract I assume you mean 45-75 rather than 45+-75?	Typo revised	abstract line 11
The results section of the abstract presents too many results without enough context, I'd suggest taking out some of the cost/step and focussing on the long-term results.	Revised as suggested	abstract lines 26-27
It might be worth reflecting more on the limited geographic area your analysis covered. How does	We have analysed nationally representative dataset and added the following text.	

Reviewer's comments	How addressed	Location of revision
this area compare nationally in terms of fitness/obesity/wealth levels.	Compared with national averages (from Health Survey for England 2012 dataset) for the same age range of the PACE-UP trial, the trial sample were more overweight/obese (66% vs 61%), more likely to have/have had a higher managerial, administrative, professional occupation (59% vs 36%), and less likely to be white (80% vs 93%).	Page 4 (second paragraph)
line 20 of page 4 I assume you mean 'It is shown' rather than 'It is shows'	Revised as suggested	Page 4 (second paragraph)
line 28 p4 you are critical of the generalizability of other studies but do you feel yours is highly generalizable?	Thank you, this is an important point and one which the trial investigators have considered specifically. We have added the following text: Other challenges relate to the generalisability of effectiveness data, given the focus on South London and 10% recruitment rate, even though recruitment was comparable with other PA trials^{33, 34}. The trial was shown to recruit fewer: men, people aged 55-64 compared with those over 65yrs, people from the most deprived quintile compared with least deprived, and Asian compared with white people³⁵. However, there was good representation of women, older adults and people who were overweight, all of whom are groups likely to benefit from the intervention⁴. Investigation into the reasons for non-participation showed an important minority cited existing medical conditions, too many other commitments or considered themselves to be sufficiently active^{35, 36}.	Page 13 (second paragraph)
line 17 p5 'two' rather than 'wo' and 'against a control'. Overall the paper really needs a good proof read, it was a bit disappointing to see so many	Apologies, the typo has been amended, proof-reading improved and text revised accordingly.	Page 5 (fourth paragraph)

Reviewer's comments	How addressed	Location of revision
errors that even a word processor should have picked up.		
p6 you talk about the missing data but what was the impact of your imputation, I am happy for it to be the base-case but it would be good to know what the scale of missingness was and something of the impact of imputation. Did you follow a good practice guide e.g. Faria et al (2015)?	The impact of missing data has been explored via sensitivity analysis. The findings were not decisional. More detail is given in the methods and results sections respectively covering the methods of accounting for missing data and the extent of missingness. Please find added text below. Text in methods section now reads as: Standard practice for accounting for missing data was followed.^{19, 20} Patterns of missing data were investigated, with multiple imputation by chained equations fitted to replace item non-response. Missing EQ-5D data were replaced using an index rather than domain imputation as recommended²¹. Mean imputation was used where missing data was $\leq 5\%$²². Imputation models were fitted to match the model used for main analysis whilst including the predictors of missingness as appropriate. Second, the dependent variables were included in imputation models to ensure that the imputed values have similar relationships to the dependent variable as the observed values²³. Deterministic sensitivity analyses assessed: (c) exclusion of missing data; Text added as footnote to Table S7 Pattern of missing data for the base case analysis was multivariate (i.e. some but not all variables had data missing for some participants). The amount of missing data, where observed, was less than 5% except for EQ5D scores (baseline data: 5% (n=51); 3	Page 6 (third paragraph) Page 7 (second paragraph) Footnote to

Reviewer's comments	How addressed	Location of revision
	months data: 7% (n=67); 12 months data: 8%(n=74).	Supplementary Table S7
p8 line 3, why not a full lifetime scale rather than to 88 years? Won't make much difference at all	Yes, you are correct, using a full lifetime scale won't make much difference. Our choice of 88 years was to reflect the average life expectancy of the trial cohort (aged 59 years) and was based on data from Office of National Statistics. This is currently reflected in P8 lines 21-22.	n/a
p8 line 23, '...physical activity and the interventions.' where what, included/excluded?	The included costs are provided in supplementary file Table S4. We have added the following text to improve clarity.... Scenario 3 plus all patient costs related to participation in physical activity and the interventions (details of the participants' costs are provided in supplementary file Table S4).	Page 9 (second paragraph)
It isn't very clear if the control group matches what you consider to be a 'no intervention' group, or by the time patients were enrolled are they already subject to some intervention? From a knee jerk reaction their costs seemed high in the short-term model if they are no intervention	Thank you for noting this needed clarification. The control group was not subject to an intervention and can be considered a 'no intervention group'. The published protocol (Harris et al, 2013) explained participants were asked to maintain usual PA and that there was no PA intervention. We have added text '(ie 'no intervention group') for clarity. The last sentence of the fourth paragraph on Page 5 provides details of the control group. The costs for the control group (and intervention groups too) in the short-term cost-effectiveness analysis were health service use (see Table 1 column 6). These were included following good practice guidance on accounting for the costs consequences of	Page 5 (fourth paragraph)

Reviewer's comments	How addressed	Location of revision
	interventions.	
End of p8 start of p9 you need to be clearer about the costs, I didn't really follow why you would include or exclude the different costs.	We did not include or exclude different costs. The text was meant to provide further information on how the three arms compared in terms of the components of the total costs. We have revised the text to improve clarity (see below). In terms of the components of total costs, the cost of nurse-supported pedometer delivery was seven times greater (£50) than the postal group (£7), and set-up costs was double. Comparing the trial arms based on cost of health service use shows that the control group cost £35 more per participant than the postal group and £12 more than the nurse group. Results are similar at 12 months, except for the control arm, which has a higher overall average cost than the postal arm.	Page 9 (third paragraph)
Was different EQ5D at baseline adjusted for? The change over the year seems so small that I'm not really sure what to make of the results, or is that what you aimed to prove? I cant imagine that any intervention like this that is aiming for a long term impact would have a noticeably one year QoL impact.	The quality life analysis adjusted for baseline EQ5D scores, as recommended (Glick et al 2014). This is mentioned in the first paragraph of page 7. You are also correct, the significant effects of lifestyle interventions on quality of life tend to occur in the long term, which emphasises the importance of providing a long-term cost-effectiveness model.	n/a
I don't see why Figure 1 and 2 don't report the control, a CEAC isn't incremental it is raw % probability of being cost-effective at different thresholds for all interventions. Why are you only reporting for 2 interventions? Were these results used for your headline % CE?	For brevity and clarity, we presented the CEACs for the 2 interventions, with each compared to the control. Decisions makers are likely to be more interested in finding out how the new programmes (interventions) compares with the control. This is currently reflected in the existing text (p10 line 26 – P11 line 2) and titling of each Figure.	n/a
You should reflect more that the patient group were self-selected,	The following text has been added to reflect	

Reviewer's comments	How addressed	Location of revision
this is very unlikely to anywhere near as effective in a more general population.	this: Other challenges relate to the generalisability of effectiveness data, given the focus on South London and 10% recruitment rate, even though recruitment was comparable with other PA trials^{33,34}. The trial was shown to recruit fewer: men, people aged 55-64yrs compared with those over 65yrs, people from the most deprived quintile compared with least deprived, and Asian compared with white people³⁵. However, there was good representation of women, older adults and people who were overweight, all of whom are groups likely to benefit from the intervention⁴. Investigation into the reasons for non-participation showed an important minority cited existing medical conditions, too many other commitments or considered themselves sufficiently active^{35, 36}.	Page 13 (last paragraph)
A brief comment on why the total QALYs in table 3 are the same across groups but incremental are different would be helpful. I assume it is a rounding issue, in which case you might want to report more decimal places. I also find total QALYs easier to interpret if reported as a per patient average but that might be personal preference only.	Thank you for pointing this out, we have reported more decimal places for QALYs.	Table 3 (fifth column)

Reviewer: Robert Sloan

Reviewer's comments	How addressed	Location of revision
---------------	----------------------

Reviewer's comments	How addressed	Location of revision
The analysis must be redone not using > 10minute bouts. The guideline has changed. Please read https://emea01.safelinks.protection.outlook.com/?url=https%3A%2F%2Fhealth.gov%2Fpaguidelines%2Fsecond-edition%2Freport%2Fpdf%2F02_A_Executive_Summary.pdf&data=01%7C01%7Cjulia.fox-..... Bouts, or episodes, of moderate-to-vigorous physical activity of any duration may be included in the daily accumulated total volume of physical activity. The 2008 Physical Activity Guidelines for Americans recommended accumulating moderate-to-vigorous physical activity in bouts of 10 minutes or more. Research now shows that any amount of moderate-to-vigorous physical activity counts toward meeting the target range. Previously, insufficient evidence was available to support the value of bouts less than 10 minutes in duration. The 2018 Committee was able to conclude that bouts of any length contribute to the	Many thanks for the recommended literature, 2018 Physical Activity Guidelines Advisory Committee Scientific Report In our trial, the “analyses of total MVPA as the outcome produced almost identical effect size estimates as found with MVPA in >=10-min bouts; at 12 mo, postal versus control was 36 (95% CI 17, 55) min/wk and nurse versus control was 32 (95% CI 13, 50) min/wk. In other words, all of the increase in MVPA was in >=10-min bouts” (Harris et al 2017; http://journals.plos.org/plosmedicine/article?id=10.1371/journal.pmed.1002210) When the trial was performed, the guidelines related to MVPA in >=10-min bouts and this is what is published as main results in the main trial paper (Harris et al 2017). In fact, UK guidelines still currently relate to 10 min bouts. Whilst we recognise the advisory committee scientific report is a recommendation to CDC/US government, the new US guidelines have not been published yet (although it is likely to account for this recommendation). We have therefore retained the current focus of analysis.	n/a

Reviewer's comments	How addressed	Location of revision
health benefits associated with the accumulated volume of physical activity		

VERSION 2 – REVIEW

REVIEWER	Sebastian Hinde Centre for Health Economics, University of York
REVIEW RETURNED	11-Jun-2018

GENERAL COMMENTS	The authors have thoughtfully and thoroughly dealt with the comments I raised. As a result I am happy to recommend this paper for publication. Congratulations on a good piece of work.
---

REVIEWER	Robert SLoan PhD Kagoshima University Graduate Medical School
REVIEW RETURNED	16-Jun-2018

GENERAL COMMENTS	How was objectively measured MVPA by accelerometer calculated/determined? Please include in methods and cite.
---

VERSION 2 – AUTHOR RESPONSE

Reviewer 1: We are happy that you find the changes acceptable. Thank you for your review, helpful comments and kind remarks on our paper.

Reviewer 2

- We have added " Physical activity was measured objectively by accelerometry (GT3X+. Actigraph LLC) and data were reduced using Actilife software (v 6.6.0). The summary variables used were as follows: step-counts; and time spent in MVPA in ≥ 10 minute bouts ($\geq 1,952$ Counts Per Minute, equivalent to ≥ 3 Metabolic Equivalents. 17" to page 6 (lines 11-14)

- We have deleted "based on objectively measured PA by accelerometer" from p6 lines 16-17

- We have added a new reference (no17) "Freedson PS, Melanson E, Sirard J. Calibration of the computer science and applications, Inc. accelerometer. Med Sci Sp Exe. 1998;30:777-781", and amended the reference numbering that follow within the reference list and within the text.

In addition to the comments from reviewers, we have improved the standardisation and information given in the references and corrected the order of referencing. We have added a paragraph in about patient and public involvement in the study and supplied evidence for this with further referencing..

I have resubmitted the clean version and copy with changes marked. As none of the other documents required changes, I have not resubmitted these. Please let me know if you need anything else.